# Evaluating the Performance of a Magnetic Nanoparticle-Based Detection Method Using Circle-to-Circle Amplification

**DOI:** 10.3390/bios11060173

**Published:** 2021-05-28

**Authors:** Darío Sánchez Martín, Reinier Oropesa-Nuñez, Teresa Zardán Gómez de la Torre

**Affiliations:** 1Division of Nanotechnology and Functional Materials, Department of Material Sciences and Engineering, Ångström Laboratory, Uppsala University, 751 03 Uppsala, Sweden; dario.sanchez@angstrom.uu.se; 2Division of Solid-State Physics, Department of Material Sciences and Engineering, Ångström Laboratory, Uppsala University, 751 03 Uppsala, Sweden; reinier.oropesa@angstrom.uu.se

**Keywords:** DNA detection, magnetic nanoparticles, Brownian relaxation, circle-to-circle amplification, nanoparticle functionalization, zoonotic diseases

## Abstract

This work explores several issues of importance for the development of a diagnostic method based on circle-to-circle amplification (C2CA) and oligonucleotide-functionalized magnetic nanoparticles. Firstly, the performance of the detection method was evaluated in terms of sensitivity and speed. Synthetic target sequences for Newcastle disease virus and Salmonella were used as model sequences. The sensitivity of the C2CA assay resulted in detection of 1 amol of starting DNA target with a total amplification time of 40 min for both target sequences. Secondly, the functionalization of the nanoparticles was evaluated in terms of robustness and stability. The functionalization was shown to be very robust, and the stability test showed that 92% of the oligos were still attached on the particle surface after three months of storage at 4 °C. Altogether, the results obtained in this study provide a strong foundation for the development of a quick and sensitive diagnostic assay.

## 1. Introduction

The spread of infectious diseases that emerge or re-emerge at the interfaces between animals, humans, and their ecosystems is becoming a large global problem. It has been identified by the WHO that 60% of known human infectious diseases have their source in animals (whether domestic or wild) [1,2]. It is therefore crucial to perform early diagnosis and efficient monitoring of pathogen spread. Newcastle disease and salmonellosis are two examples of zoonotic diseases. Newcastle disease, caused by Newcastle disease virus (NDV), has been a very serious problem for poultry production in many countries, where outbreaks can cause flock mortality approaching 100% in fully susceptible chickens [3]. Salmonella, in the other hand, is a bacterial pathogen, mainly food-borne, that affects the intestinal tract. In Europe, salmonellosis is the second most common foodborne disease after campylobacteriosis [4]. Conventional diagnosis of pathogens is generally carried out by isolation and cultivation of virus and bacteria, which is time-consuming and laborious. It is therefore essential to use molecular-based diagnostic methods for faster detection of pathogens. These methods are sensitive enough and can be performed within a few hours. In recent years, isothermal DNA amplification techniques have gain widespread use in many applications due to their simplicity compared to polymerase chain reaction (PCR) method. One of such techniques is the rolling circle amplification (RCA), which produces a long single-stranded DNA molecule with repeating sequences of a circular padlock probe. Padlock probes are synthetic single-stranded oligonucleotides that contain two target-specific regions at the 3′ and 5′ ends, as well as a linker region that may contain specific sequences for purposes such as detection or restriction enzyme digestion [5]. Upon target recognition and hybridization, the ends of the padlock probes are brought into juxtaposition, allowing for circularization by a ligation enzyme. The ligation procedure is very specific and a single mismatch near the ligation junction prevents ligation [6]. Amplification of the circularized molecules is achieved through RCA [7,8]. In order to increase the sensitivity of the assay, one can employ the circle-to-circle amplification (C2CA) method, an extension of RCA [9]. The concatemers obtained during RCA are digested at a specific restriction site, re-ligated into new padlock probe circles, and further amplified by an additional RCA. In this way, the number of amplified products (also denoted as RCA products) increase exponentially. The C2CA method has previously been used for detection of Ebola, Zika fever, and severe acute respiratory syndrome coronavirus 2 [10,11].

Several methods for detecting and analyzing RCA products have been reported, such as the use of dye conjugated deoxyribonucleotide triphosphate (dNTP), fluorescent probes, and different types of nanoparticles [12,13]. In this work, magnetic nanoparticles were used to confirm the presence of RCA products. Magnetic nanoparticle-based biosensors offer an attractive and cost-effective route for detection of biomolecules since magnetic nanoparticles are relatively inexpensive to produce. In comparison to fluorophore tags, which are light-sensitive and therefore require special handling, magnetic nanoparticles are both physically and chemically stable [14,15,16,17]. Furthermore, the read-out instrumentation can be produced inexpensively, as it contains no optics but only cheap electromagnetic device components. The Brownian relaxation detection principle is a substrate-free detection method where suspended magnetic nanoparticles, exhibiting Brownian relaxation behavior, are functionalized with probe molecules for recognition of specific analytes [18,19]. Binding of the analyte to the probes causes a hydrodynamic size increase of the nanoparticles, corresponding to the size of the target molecule. This causes a decrease in the relaxation frequency in the imaginary part of the complex magnetization spectrum of the particles since the position of the peak is inversely proportional to their hydrodynamic size. The amount of analyte in the sample can also be monitored as a corresponding decrease of the amplitude of the relaxation peak of the non-bound particles [20]. We have developed a detection approach were the padlock probe and RCA technology is combined with the Brownian relaxation detection principle, called the volume amplified magnetic nanobead detection assay (VAM-NDA) [20]. Here, single-stranded oligonucleotides (also denoted as oligos) are complementary to the repeating sequences in the RCA products. It has previously been shown that the VAM-NDA sensitivity was estimated to be 5 pM (1 fmol in 200 µL) of RCA product using one single amplification step [21,22]. The current work explores how the performance of the assay is improved using the C2CA method in terms of sensitivity and assay speed. Synthetic NDV and *Salmonella* targets were used as model sequences for this study where different DNA amplification times were analyzed. The performance of the assay was also analyzed with biological samples spiked with synthetic NDV target. Finally, the oligo functionalization on the nanoparticle surface was evaluated in terms of robustness and stability. This is of great importance since the quality and stability of the oligo functionalization may affect the performance of the bioassay.

## 2. Materials and Methods

Sequences of the used targets, padlock probes, and detection oligos are listed in Appendix A.

### 2.1. Conjugation of Detection Oligos to Magnetic Nanoparticles

The magnetic nanoparticles used in this study consist of a core made up of a magnetic nanoparticle aggregate, encapsulated with hydroxyethyl starch. The core consists of 75–80% (*w*/*w*) magnetite according to the manufacturer (Micromod Partikeltechnologie GmbH, Rostock, Germany). The particles are streptavidin-functionalized and have a particle diameter of 100 nm and a concentration of 10 mg/mL. For the functionalization with biotinylated oligos, the nanoparticles were washed three times with a washing buffer (1 × Wtw buffer) consisting of 10 mM Tris-HCl, 5 mM EDTA, 0.1% Tween 20, and 0.1 M NaCl using a magnetic separator (SuperMag Multitube Separator, Ocean Nanotech, San Diego, CA, USA). The nanoparticles were thereafter resuspended in 1 × Wtw buffer in half the original volume. The desired amount of biotinylated single-stranded oligos (biomers.net, Ulm, Germany) were added to the nanoparticle solution, and the mixture was incubated at room temperature for 20 min. In order to remove unreacted oligos, the particles were washed three times with 1 × Wtw buffer, and thereafter resuspended in 1 × PBS to their original volume. The average number of detection oligos per nanoparticle was estimated by comparing the fluorescence of the oligo-coupled magnetic nanoparticles to a dilution series containing free fluorescent oligos and non-functionalized nanoparticles. The emission scans (excitation: 470 nm, and emission: 505–560 nm) were performed using a fluorescence spectrophotometer (Infinite^®^ 200, Tecan, Stockholm, Sweden).

Four oligo/nanoparticle ratios were analyzed (25:1, 50:1, 75:1, and 100:1). Thereafter, the average number of oligos per nanoparticle was estimated by fluorescence spectroscopy as described above. Here, the NDV C2CA oligos were used for the experiment. To evaluate the stability of the functionalization, we functionalized the nanoparticles with an oligo/nanoparticle ratio of 50:1 with the NDV C2CA detection oligo. Thereafter, the functionalized particles were covered and stored either at room temperature or at 4 °C for six months. The fluorescence of each sample was measured, as described above, directly after functionalization, three and six months after. The NDV C2CA oligos were also used in this case. There were triplicates for all sample types.

### 2.2. Target Recognition, Ligation, and RCA

The ligation and amplification procedure for producing a stock solution containing 0.5 pmol (in 100 µL) RCA products has been reported earlier [23]. Briefly, a ligation mixture containing 3 pmol DNA target, 1 pmol phosphorylated padlock probe (biomers.net), 1 mM ATP (Thermo Fisher Scientific, Stockholm, Sweden), 20 mU/µL T4 ligase (Thermo Fisher Scientific), and 1 × phi29 buffer (Thermo Fisher Scientific) was incubated for 15 min at 37 °C in order to let the padlock probe hybridize to the target and form a molecule circle through ligation. Thereafter, the ligation mixture consisting of 1.2 pmol of circles was mixed with 167 µM dNTP (Thermo Fisher Scientific), 0.2 µg/µL bovine serum albumin (BSA; Thermo Fisher Scientific), 2.7 mU/µL phi29 DNA polymerase, and 1 × phi29 buffer, and the reaction was incubated at 37 °C for 60 min followed by enzymatic inactivation at 65 °C for 5 min. Hybridization buffer (0.1 M Tris-HCl (pH 8), 0.1 M EDTA, 0.5% Tween-20, 2.5 M NaCl) was then added to the RCA mix to obtain the desired RCA product concentration. Duplicates samples of 0.5 pmol RCA products were made. Samples containing 1, 10, and 100 fmol of RCA products were prepared through mixing appropriate amounts of the RCA samples with hybridization buffer. Water was used instead of DNA target in the ligation mixture for the negative control (NC) samples.

### 2.3. Target Recognition, Ligation, and C2CA

Samples from a dilution series (0, 1, 10, and 100 amol of target) were mixed together with a ligation mixture containing 2 pmol phosphorylated padlock probes (biomers.net), 0.2 µg/µL BSA, 250 mU/µL Tth DNA ligase (GeneCraft, Köln, Germany), and 1 × Tth ligase buffer (GeneCraft), and were incubated for 5 min at 60 °C. In the case of the *Salmonella* samples, the ligation mixture also contained a capture oligonucleotide (1 pmol) complementary to part of the target. The interacting DNA molecules (target, padlock probes, and capture oligo in the case of *Salmonella*) were captured by My One T1 Dynabeads (Thermo Fisher Scientific) through a biotin group by rotating the sample for 10 min at room temperature. Biotin was present in the NDV target and in the *Salmonella* capture oligo. After incubation, the Dynabead particles were washed once with 1 × Wtw buffer using a permanent magnet to eliminate any unreacted padlock probes that may interfere with downstream reactions. The circles formed in the ligation reaction were amplified by the addition of 20 µL RCA mixture containing 0.2 µg/µL BSA, 125 µM dNTP, 100 mU/µL phi29 DNA polymerase, and 1 × phi29 buffer, and the reaction was incubated at 37 °C for either 20 or 60 min followed by enzymatic inactivation at 65 °C for 1 min. To generate new sequences for the second amplification round, the RCA products were digested by adding a mixture containing 3 pmol replication oligos, 0.2 µg/µL BSA, and 40 mU/µL *AluI* (Thermo Fisher Scientific) in 1 × phi29 buffer. The reaction was incubated for 1 min at 37 °C, and *AluI* was inactivated by 1 min at 65 °C. The monomer products produced in the digestion step were isolated from the Dynabead particles using a permanent magnet and mixed with a solution containing 0.2 µg/µL BSA, 0.67 mM ATP, 100 µM dNTP, 14 mU/µL T4 DNA ligase, and 60 mU/µL phi29 DNA polymerase in 1 × phi29 buffer. The samples were thereafter incubated for either 20 or 60 min at 37 °C, allowing the monomers to be ligated and amplified. The reaction was inactivated by 1 min at 65 °C.

For the biological samples spiked with synthetic NDV target, the C2CA protocol described above was implemented. In addition, 100 amol of plasmid pUUH239.2 were added in the ligation reaction for all samples. For that, a stock of plasmid was previously digested with restriction enzymes *AluI* and BsuRI(HaeIII) (Thermo Fisher Scientific) for 1 h (1 × Tango Buffer, 4000 attomoles of plasmid, 175 mU/µL *AluI*, and 350 mU/µL BsuRI). The digestion was carried out for 1 h at 37 °C and followed by 10 min of inactivation at 65 °C. Immediately before the ligation for the C2CA reaction the plasmid DNA was denaturalized through heat by incubating it for 5 min at 96 °C. The plasmids used in this work originated from *Klebsiella pneumoniae* and have been harvested from *Escherichia coli.*

All samples were made on duplicates, except for the biological samples, where they were made on triplicates.

### 2.4. AC Susceptibility Measurement on Samples Containing RCA Products and Probe-Tagged Magnetic Nanoparticles

Fifty microliters of sample containing RCA products and 20 µL functionalized magnetic nanoparticles (2 mg/mL, 50 DO/particle) were mixed and incubated for 20 min at 55 °C. For NDV RCA products produced as in Section 2.2, 20 µL of sample was mixed with 20 µL functionalized magnetic nanoparticles and thereafter incubated for 20 min at 55 °C.

The frequency-dependent magnetic susceptibility was measured at room temperature using an AC susceptometer (DynoMag^®^, Acreo, Gothenburg, Sweden) in the frequency range of 5–50,000 Hz with 20 frequency points. To account for variations in magnetic material between samples, the magnetic response for each sample was normalized with respect to the amount of magnetic material as described in previous works [23]. In summary, the data were divided with a constant value in the in-phase component of the volume susceptibility (χ’_inf_), as this value was proportional to the total amount of magnetic material in a sample. In this work, all data were normalized against the χ’_inf_-value at 4400 Hz. The χ”-values and corresponding standard deviation were calculated for all samples.

### 2.5. Determination of the Limit of Detection (LOD)

In this study, the LOD was defined as the lowest tested amount of DNA target yielding a magnetic response that differed by more than three standard deviations from the magnetic response of the negative control (sample without target DNA).

## 3. Results and Discussion

To investigate the influence of the quantity of oligos in the functionalization, we studied four different oligo/nanoparticle ratios (25:1, 50:1, 75:1, and 100:1). The fluorescence intensity for the different oligo amounts is presented in Figure 1a. The emission value scaled almost linearly with the oligo/nanoparticle ratio, and the small error bars for each data point highlighted the small batch-to-batch variation between the samples, making the functionalization procedure very robust. The presented results show that the used MNPs can bind at least 100 detection oligos on average.

The oligo functionalization was evaluated over time at two different storage temperatures (Figure 1b). Here, the nanoparticles were functionalized with an oligo/nanoparticle ratio of 50:1. Immediately after the functionalization, the number of oligos per nanoparticle was estimated to be 49 ± 3 (*n* = 6) by comparing the fluorescence of the oligo-coupled magnetic nanoparticles to a dilution series containing free fluorescent oligos and non-functionalized nanoparticles. The results showed that the loss of oligo surface coverage on the particles when these were stored at 4 °C for three months was about 8%, while in the case of the storage at room temperature for three months, the loss was around 30%. These results are in good agreement with previous results where similar oligo detachment was seen for similar nanoparticles (100 nm BNF particles-avidin, 10 mg/mL, Micromod) [24]. The total loss increased to 23% and 38% of oligos from the nanoparticle surface for the samples stored in the fridge (4 °C) and at room temperature, respectively, after six months of storage. To avoid major detachment of oligos from the nanoparticles surface, the functionalized nanoparticles should preferably be stored at 4 °C for no more than three months.

The χ”_max_ values of samples containing different amounts of NDV RCA products (1, 10, and 100 fmol) are presented in Figure 2. These values were obtained from the imaginary part of the complex susceptibility curves displayed in Appendix A. As expected, the normalized χ”_max_ value decreased with increasing amount of RCA products, indicating that the number of bound nanoparticles increased when a sample contained more RCA products. These results show that it is possible to differentiate a sample containing 1 fmol of RCA products from its NC using an RCA time of 60 min, which is in good agreement with previous studies [21,22]. Moreover, there is no significant difference between the NC sample and the blanks (suspended nanoparticles in PBS), which ensures that there is unspecific binding of the MNPs in the NC and it can therefore be a reliable NC sample.

To achieve higher sensitivity, we employed the C2CA technique. A schematic illustration of the working principle is presented in Figure 3. The assay started with hybridization of a padlock probe to a target sequence, and a DNA ligase circularized correctly matched padlock probes. The biotinylated targets were thereafter bound to streptavidin-coated Dynabead particles and amplified with RCA if the target had bound a circularized padlock probe. The amplification products were then digested to monomers by a restriction endonuclease enzyme that cuts at a defined sequence motif. The monomers were thereafter ligated and amplified by another round of RCA to produce a new set of RCA products. The new set of RCA products was detected by the hybridization of oligo-functionalized MNPs.

Three NDV target amounts (1, 10, and 100 amol) and a NC sample with no NDV target were analyzed. The target was diluted in steps of 1:10 and processed according to the C2CA protocol as described in the Materials and Methods section. The two amplifications steps were conducted for either 20 or 60 min in order to understand how the amplification time influences the performance of the assay. Four different types of samples were analyzed, on the basis of their amplification times, namely, 20′+20′, 20′+60′, 60′+20′, and 60′+60′, where the first number indicates the amplification time in first amplification step and the second number indicates the amplification time in the second amplification step. The results are presented in Figure 4 where the mean normalized χ”_max_ values at 145 Hz are plotted as a function of DNA target amount. The 145 Hz frequency corresponded to the Brownian relaxation peak of the unbound nanoparticles. All χ”_max_ values are extracted from the magnetization spectra in Appendix A.

All measured samples were detectable for all the amplification times. For the 1 amol samples, there was a peak decrease of 5, 15, 5, and 36% compared to the appropriate NCs in the cases of 20′+20′, 20′+60′, 60′+20′, and 60′+60′, respectively. Employing 20 min of amplification in the second RCA resulted in the same peak decrease, independently of the amplification time used in the first RCA. This indicated that the size of the RCA products may have had a greater impact on the performance of the assay compared to the number of templates in a sample.

The 1 amol samples with the highest amplification time in the second RCA (60 min) had the greatest decrease in amplitude. This was particularly noticeable in the case of 60′+60′. Such a result was expected since these samples contained more and larger RCA products compared to the other 1 amol samples.

In the case of the 10 amol samples, the peak values were close to 0 for 20′+60′ and 60′+60′, indicating that the majority of the functionalized nanoparticles were bound to the RCA products. The 60′+20′ samples showed somewhat higher values compared to 20′+60′ and 60′+60′, indicating that there was a slightly higher number of free nanoparticles in those samples. This demonstrated that the size of the RCA products may have had an important impact on the performance of the assay.

For the samples with 100 amol of target, the amplitude values at 145 Hz seemed to increase with total amplification time. However, these values did not represent a peak or local maximum in the magnetization spectra (Appendix A), but were instead the tails of peaks seen at lower frequencies. Lower frequency peaks corresponded with higher hydrodynamic radius [25,26]. In this case, it could be due to the arrangement of clusters of small particles or binding of nanoparticles to smaller, less amplified RCAs. Nevertheless, the values at 145 Hz found for these samples most likely did not represent unbound nanoparticles.

To evaluate if it was possible to extrapolate these results to another type of DNA target sequence, we used the same C2CA protocol but with a synthetic *Salmonella* DNA. In this case, only 20′+20′ of amplification was employed. As 20+20 min of amplification yielded detectable drops of amplitude in all samples with NDV DNA, it was the only amplification time used for *Salmonella*. A similar outcome can be observed in Figure 5 for the *Salmonella* samples where a quantitative response was achieved with a limit of detection of 1 amol.

As a final step in this work, the performance of the assay was evaluated by measuring biological samples containing 100 amol plasmids spiked with different amounts of synthetic NDV target (Figure 6). For this, 20′+20′ of amplification time was used. The results showed that there was a small difference between the negative control and the 1 amol sample, but according to the definition of the LOD, there was no significant difference between the samples, indicating that the sensitivity of the assay was reduced to somewhere between 1 and 10 amol. The reduction in sensitivity can be deduced to the increased amount of plasmid ssDNA present in the samples. This presence plasmid could have had some effect in the target recognition step where binding between the plasmid and the target sequence can occur due to complementary regions in the plasmid sequence. This will in turn reduce the quantity of circular target–padlock probe complexes in the samples that can further be amplified.

Table 1 presents several DNA detection methods for NDV and *Salmonella.* The LOD of our method was somewhere between 1 and 10 amol (5–50 fM in 200 µL) using an assay time of about 80 min (40 min C2CA, 20 min labeling, and 20 min readout). This made our detection method more sensitive than most of the presented *Salmonella* assays. The detection method based on target recycling amplification has about 70 times better sensitivity but lacks in detection speed. In the case of NDV detection, the LAMP assay based on MNPs is much more sensitive and rapid compared to the proposed method, but LAMP has the disadvantage of carryover contamination [27,28] and the appearance of false-positive results caused by the unwanted secondary structures formed by the primers.

Overall, using C2CA with MNPs provides an increased sensitivity compared to many other methods while still offering a read out-time of less than 1.5 h. Compared to more sensitive methods such as PCR and LAMP, C2CA offers a decreased risk of contamination. Moreover, C2CA in combination with MNPs relies on stable reagents that are easy to handle, compared to fluorophores used for PCR and LAMP. Here, we show that C2CA worked well, even with a large quantity of plasmid DNA in the sample. This makes it a robust method for POC diagnostics in health centers lacking experienced lab personnel or labs with sterile conditions.

## 4. Conclusions

The results obtained in this work highlight the strengths of a magnetic nanoparticle-based biosensor in combination with C2CA, which could work well for detection of pathogens and zoonotic diseases. The magnetic nanoparticles used in this work showed robust functionalization over time, with small batch-to-batch variation. These facts put them ahead of fluorophore-based detection methods, as they kept reliably in a fridge for at least three months without a significant loss of bound oligos. C2CA was employed with a total of 40 min of amplification (20′+20′), which yielded three orders of magnitude better LOD than 60 min of normal RCA for both reference sequences. The assay was also evaluated with biological samples spiked with synthetic NDV, and it was shown that the sensitivity of the assay decreased. While there are things to improve, such as simplifying the method or producing a kit, our method had the advantage of being inexpensive, rapid, and robust. All of these properties make our method suitable for developing a test for the point-of-care.

## Figures and Tables

**Figure 1 biosensors-11-00173-f001:**
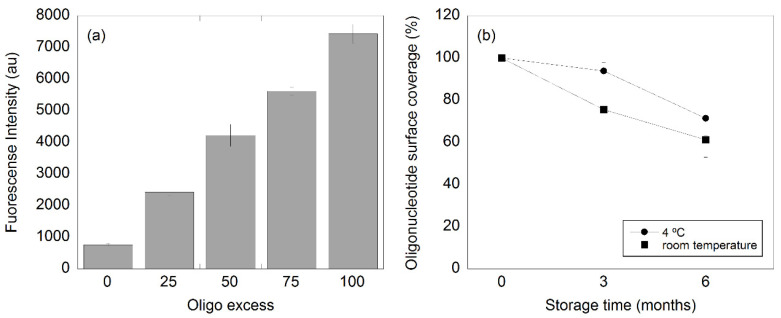
(**a**) Fluorescence intensity with respect of the oligo/nanoparticle ratio used in the nanoparticle functionalization and (**b**) oligo detachment from the nanoparticle surface over time (three and six months). The particles were stored at either room temperature or in the fridge at 4 °C.

**Figure 2 biosensors-11-00173-f002:**
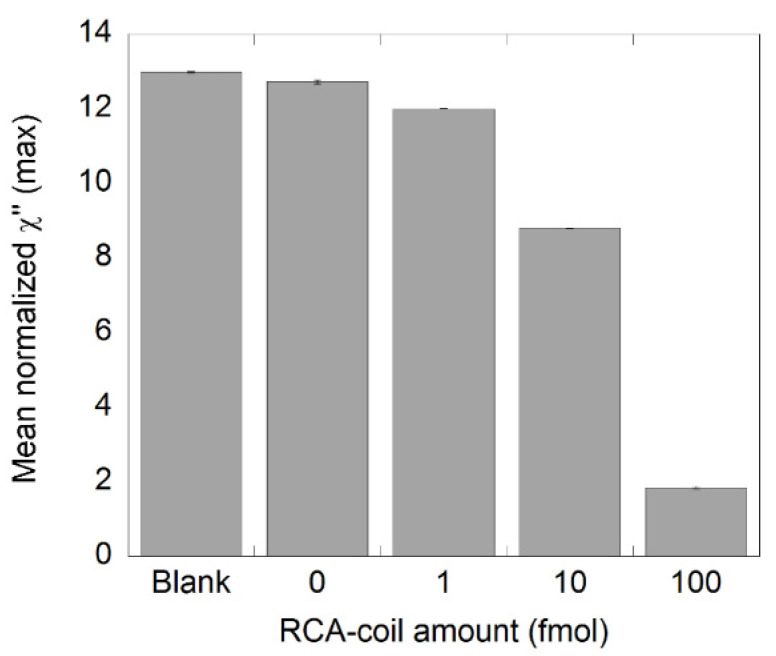
Detection of synthetic NDV DNA diluted in steps of 1:10 using one round of RCA. A 0.5 pmol RCA product stock solution was diluted to the desired amounts and measured in the DynoMag. The plot shows the number of RCA products and the corresponding normalized χ”_max_ values, representing unbound nanoparticles. Error bars represent standard deviation based on two independent samples.

**Figure 3 biosensors-11-00173-f003:**
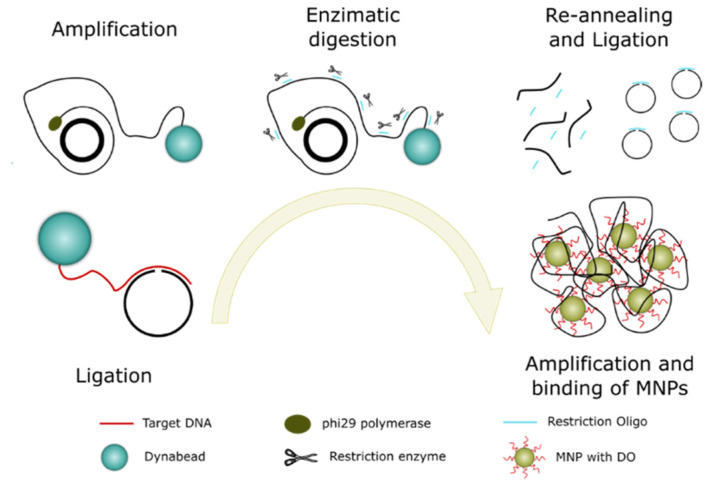
Schematic representation of the C2CA protocol and the different steps involved.

**Figure 4 biosensors-11-00173-f004:**
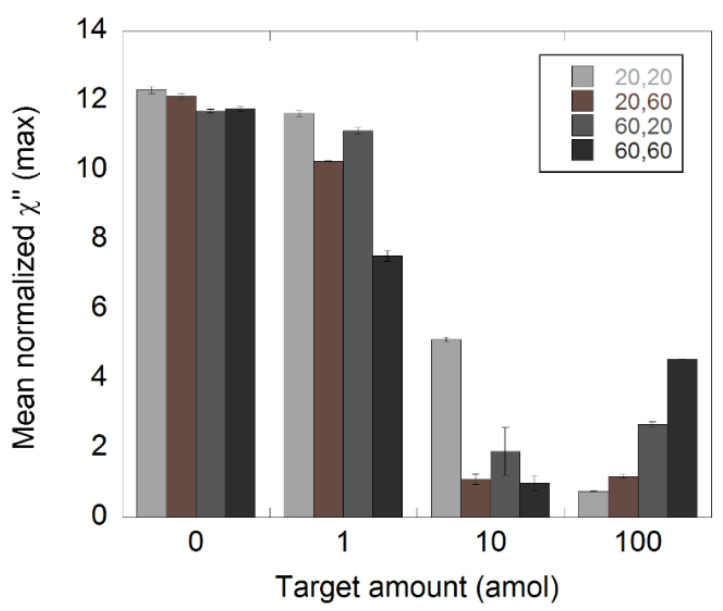
Detection of synthetic NDV DNA diluted in steps of 1:10 using C2CA. The target was diluted and processed according to the C2CA protocol. The plot shows the amount of target and the corresponding normalized χ”_max_ values, representing unbound nanoparticles for different amplification times. The standard deviations were based on measurements of two independent samples.

**Figure 5 biosensors-11-00173-f005:**
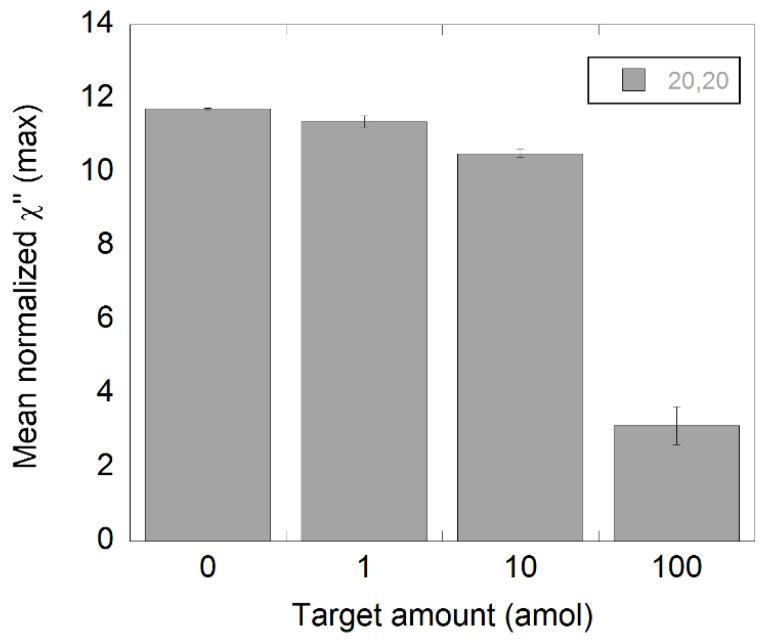
Detection of synthetic *Salmonella* DNA diluted in steps of 1:10 using C2CA. The target was diluted and processed according to the C2CA protocol. The plot shows the amount of target and the corresponding χ”_max_ values, representing unbound nanoparticles. The standard deviations were based on measurements of two independent samples.

**Figure 6 biosensors-11-00173-f006:**
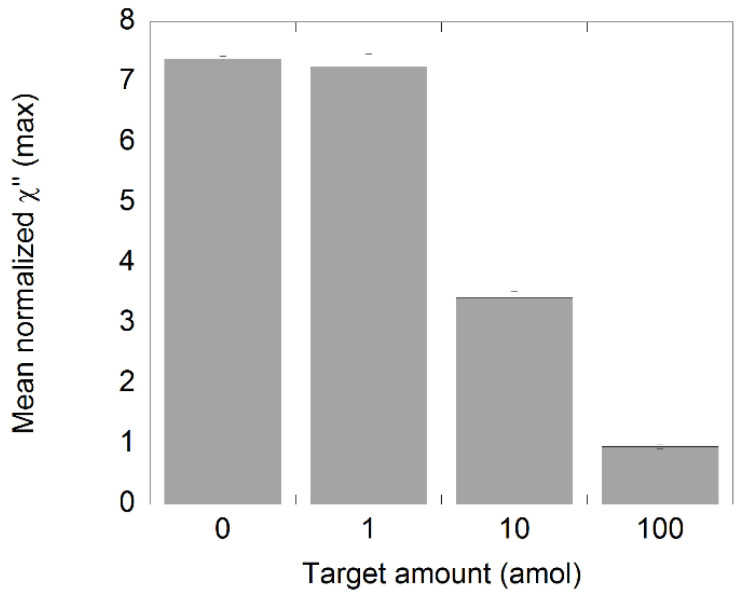
Analysis of NDV DNA spiked samples. The synthetic target was diluted and processed according to the C2CA protocol. The plot shows the amount of target and the corresponding χ”_max_ values. The standard deviations were based on measurements of three independent samples.

**Table 1 biosensors-11-00173-t001:** Different DNA detection methods for NDV and Salmonella.

Detected Pathogen	Detection Method	LOD	Assay Time	References
NDV	LAMP with MNPs	10 aM	30 min	[29]
NDV	C2CA with fluorophore tagged oligos	5–10 virus copies	-	[30]
NDV	Real time PCR with SYBR Green	10 pg DNA	-	[31]
*Salmonella*	HAD with ITO electrodes	2.4 nM	-	[32]
*Salmonella*	Target recycling amplification with gold electrode	0.67 fM	About 3 h	[33]
*Salmonella*	Target DNA oligo with QDs	4 nM	20 min	[34]
*Salmonella*	Dy labeled target DNA with MOF material	28 pM	-	[35]
NDV and *Salmonella*	C2CA with MNPs	In between 5 and 50 fM	80 min	This work

## Data Availability

Not applicable.

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
