# Peer review of "Evaluating the Performance of a Magnetic Nanoparticle-Based Detection Method Using Circle-to-Circle Amplification"

_biosensors, 2021, doi:10.3390/bios11060173_

Round 1
Reviewer 1 Report
The authors report an approach for a sensitive detection of Newcastle disease virus and Salmonella based on a circle-to-circle amplification (C2CA) and oligonucleotides-functionalized magnetic nanoparticles. This method could be achieved within an appropriate incubation time while need no sophisticated devices and the detection limit was low enough. Although the authors showed us a promising method to detect the viruses, no real clinical samples were used to validate the assay, which in my opinion is critically important. I will highly recommend the author to test some real samples to further verify the method. Therefore, I think the paper still need a more experimental data revision before it can be published in Biosensor’s journal.
- I suggest that the authors compare the sensitivity and LOD obtained for detection of the pathogen in buffered samples vs. those that are representative of practical diagnostic samples (e.g., detection in saliva, mucus, stool matrices, etc.).
- Why did you choose Newcastle disease virus and Salmonella?
- How about the selectivity results? Is proposed method is selective?
- How the authors claim the LOD of 1 aM? From Figure 5, it looks the results at 0,1 and 10 aM pathogen concentrations are same. Need statistical analysis to prove the proposed LOD.
- The authors should include the sequences of the primer, padlock probe, and target nucleic acid.
- How about an optimized incubation or reaction time (including 0 min) for an amplication product?
- I found following article for viruses’ detection based on RCA. They mentioned the LOD about 0.7 aM. https://doi.org/10.1016/j.bios.2021.113005 . You should mention the most relevant recent research in your article.
- Have the authors done any rheological characterization studies on the solutions/gels formed by the RCA reaction and C2CA process? I suggest that the rheological properties of the products formed should be characterized vs. those formed in the absence of the target species to quantify the extent of difference in the product.
Reviewer 2 Report
The study "Evaluating the performance of a magnetic nanoparticle-based detection method using circle-to-circle amplification" is very well and legibly written. However, the length of the manuscript is relatively short and the number of references used is not very large. Out of a total of 29 citations, the name "Torre" is icluded in 11 references, which is an inappropriate ratio. Internet sources are not cited correctly (e.g. ref. 2). The references are not uniform (compare e.g. ref. 20 "Zardán Gómez De La Torre, T." and ref. 21 "Gómez De La Torre, T.Z.").
What readers are noticeably missing is a classic table comparing important techniques and parameters of various methods used for the detection of Newcastle disease virus and Salmonella patogenes. Also, the list of important applications of Circle-to-circle amplification (C2CA) in the form of a clear table could better show the current state of the art.
E.g. in the graph of Fig. 4 shows "Target amount (amol)". Elsewhere in the text (Page 3) it is stated "... ligation mixture consisting of 20 nM of circles was ...". At least some of the similar records seem to be confusing. Please check that the notation of the substance amount (in mol) is not confused in the text with the concentration M (which means mol/l). This can cause considerable ambiguity and make difficult to understand the meaning of the text well.
Why was the prepared method not used for pathogen detection in a real sample? Or, why not at least use a real sample spiked with synthetic target sequence for Newcastle disease virus and / or Salmonella? This part of the study should be supplemented to actually confirm how it is written that "The results obtained in this work highlight the strengths of a magnetic nanoparticle based biosensor in combination with C2CA, that could work well for the detection of pathogens and zoonotic diseases. ".
The work contains a minimum of spelling mistakes and typos, the text and images are well processed:
Page 3: Please correct symbol for degrees of Celsius in records: "... incubated at 37 ĹźC for 60 min followed by enzymatic inactivation at 65 ĹźC for 5 min." and "... incubated for 5 min at 60 ĹźC ..." and some others. Please check the whole document for this shortcoming.
According to Crossref Similarity Check, the document compliance rate is 25%. This is mainly due to the similarities in Introduction section and Methods. The Crossref Similarity Check document is attached. Authors and editors are given the opportunity to consider whether it would not be appropriate (at least in some cases) to refer to another article than to repeat the text.

Round 2
Reviewer 1 Report
All figures look blur, I recommend the authors to replace the figures with TIFF file.